# Integrative Morphological and Molecular Diagnostics for *Euseius nicholsi* and *Euseius oolong* (Acari: Phytoseiidae)

**DOI:** 10.3390/insects16090950

**Published:** 2025-09-10

**Authors:** Xiaoduan Fang, Jun Li, Syed Usman Mahmood, Nwanade Chuks Fidelis, Jianglei Meng

**Affiliations:** Guangdong Key Laboratory of Animal Conservation and Resource Utilization, Guangdong Public Laboratory of Wild Animal Conservation and Utilization, Institute of Zoology, Guangdong Academy of Sciences, Guangzhou 510260, China; junl@giz.gd.cn (J.L.); syedusmanmahmood3@gmail.com (S.U.M.); chuk.fidel@yahoo.com (N.C.F.); mengjiangleill@163.com (J.M.)

**Keywords:** Phytoseiidae, taxonomy, morphology, molecular marker, integrative approach

## Abstract

Specimens of a phytoseiid mite were collected from Bajiaozhai National Forest Park, Guilin, China, and showed close resemblance to *Euseius nicholsi* and *Euseius oolong*. Despite similarities, some morphological traits, such as body dimensions, number of dorsal solenostomes, spermathecal shape, and setae length differed. To determine whether these variations indicated distinct species or fell within natural variation, we performed molecular analysis using mitochondrial COI, 12S rRNA, and nuclear ITS markers. The genetic divergence among the unidentified specimens, *E. nicholsi*, and *E. oolong* was minimal, and all samples clustered together in a single clade in the phylogenetic tree. This suggested that they were not separate species. We therefore concluded that the unidentified specimens are *E. nicholsi*, and *E. oolong* should be considered its junior synonym. These findings underscore the value of integrating morphological and molecular tools for accurate mite identification.

## 1. Introduction

Phytoseiid mites are important natural enemies of pest mites and small arthropods [1,2,3]. Up to now, more than 2707 species and 94 genera have been described [4]. Phytoseiid mites are very tiny, about 300–600 μm in length [1]. Some morphological characteristics are very difficult to observe and their values for species discrimination are sometimes subject to discussion [5,6,7]. Furthermore, accurate observation of morphological characteristics is highly dependent on the quality of the mounting preparations. Some features, such as the number of teeth on the cheliceral digits or the presence of solenostomes, are sometimes difficult to observe, and reliable species descriptions require the examination of multiple individuals and preparations to avoid misidentification and confusion [8].

*Euseius nicholsi* (Ehara and Lee, 1971) (Acari: Phytoseiidae) was first described from a grass in Hong Kong by Ehara and Lee [9]. This species is widely distributed in China and also reported in Thailand [4]. It is a species of great economic value, having been successfully used to control *Panonychus citri* (McGregor) (Acari: Tetranychidae), *Eotetranychus kankitus* Ehara (Acari: Tetranychidae), and *Polyphagotarsonemus latus* (Banks) (Acari: Tarsonemidae) on crops such as citrus and strawberry [2]. Liao et al. [10] described a new species, *Euseius oolong* Liao and Ho, collected on tea trees (Theaceae) at the Tea Research and Extension Station (Taiwan) and on *Bauhinia purpurea* (Fabaceae) at the Guangdong Institute of Applied Biological Resources (now named the Institute of Zoology, Guangdong Academy of Sciences, IZGAS) and Sun Yat-sen University (Guangzhou, Guangdong) [10]. *E. oolong* is morphologically very similar to *E. nicholsi*, but it differs based on its cup-shaped calyx of the spermatheca, which is funnel-shaped in *E. nicholsi*. Also, there are five solenostomes in *E. oolong* but seven in *E. nicholsi*. Differences in dorsal plate, teeth number on the fixed digit, and setae length also exist between the two species [10,11].

A survey in Bajiaozhai National Forest Park (Guilin, Guangxi) revealed that several specimens were morphologically similar to *E. nicholsi* and *E. oolong*, but some differences were observed, such as a smaller body size; dorsal shield reticulation, mainly on the anterior and lateral region; the spermathecal calyx was tubular, whereas it is cup-shaped for *E. oolong*; one/two pairs of metapodal platelets; four teeth on the fixed digit; and some setae were shorter than in *E. nicholsi* and *E. oolong*. As such, two questions arose: (i) To which species does this putative *Euseius* sp. belong? Many characteristics of the genus *Euseius* are difficult to see and interpret, especially for the shape of the spermathecal apparatus and setae length on the dorsal shield [5,6,12,13,14]. (ii) Are *E. oolong* and *E. nicholsi* synonyms? Synonymy is considered a problem for diversity characterization and correct identification of natural enemies is required for efficient biological control [15]. The combination of molecular markers with morphological analyses has allowed the identification of morphologically similar species, i.e., cryptic species or various synonyms within the family Phytoseiidae [5,15,16,17,18,19,20,21,22,23,24,25,26]. Therefore, molecular methods based on mitochondrial DNA COI, 12S rRNA, and ITS nuclear markers were applied for discrimination of the *Euseius* species herein considered.

## 2. Materials and Methods

### 2.1. Species Considered

The plant-beating method was used to collect specimens, which were then transferred into vials filled with 100% alcohol. Putative *Euseius* sp. were collected from *Cyclobalanopsis glauca* (Thunb.) Oerst. (Fagaceae), *Sabina squamata* (Buch.-Hamilt.) Ant. (Cupressaceae), etc. in Bajiaozhai National Forest Park (Guilin, Guangxi, China) (E 110.720375°, N 26.221212°). Specimens of *Euseius oolong* were collected from *Bauhinia purpurea* L. (Fabaceae) at IZGAS [Institute of Zoology, Guangdong Academy of Sciences, Guangzhou, China], at the same locality as paratypes (E 113.300552°, N 23.09173°). *E. nicholsi* were collected from three host plants in two places, including the same host and place as *E. oolong* above, and also from *Eurya macartneyi* Champion (Pentaphylacaceae) and *Ficus hispida* L. f. (Moraceae) in Shaoguan, Guangdong, respectively.

### 2.2. Morphological Analysis

A total of 15 specimens of *Euseius* sp. from different host plants in Bajiaozhai National Forest Park were considered for morphological examination. Mites were mounted in Hoyer’s medium and examined, measured, and illustrated under a phase and DIC (differential interference contrast) microscope (Zeiss^®^ Axio Imager A2, ZEISS Group, Oberkochen, Germany) and image pick-up system (ZEN 2.3 (Blue edition)^©^ Carl Zeiss Microscopy GmbH, 2011). Measurements are presented in micrometers (μm).

The type material (one female holotype, collected from Chai Wan, Hong Kong, 18-X-1970 on a grass, NSMT Ac-13078) of *E. nicholsi* was checked. Type materials of *E. oolong*, including two females of paratypes (no. 2029-1, 2) from *Bauhinia purpurea* at Guangdong Institute of Applied Biological Resources Guangzhou City, Guangdong Province, 23.II.2017, China, collected by J. R. Liao, and deposited at our Institute, were re-measured and redrawn. Three newly collected *E. oolong* specimens and three newly collected *E. nicholsi* specimens after DNA extraction were retrieved and their morphological characteristics were verified.

Idiosomal seta terminology follows that of Lindquist and Evans [27] as applied to phytoseiids by Rowell et al. [28] and Chant and Yoshida-Shaul [29], respectively, for dorsum and venter; adenotaxy and poroidotaxy terminology follows that of Athias-Henriot [30]. Morphological identification of newly collected *E. oolong* specimens was based on Liao et al. [10], and morphological identification of *E. nicholsi* was based on Ehara and Lee [9]. All specimens were deposited at IZGAS. Their characteristics were observed and measured (Table 1).

### 2.3. Molecular Experiments and Data Analysis

Three specimens from each species were used for molecular experiments. Total genomic DNA was individually extracted using the HiPure Tissue DNA Micro Kit (Item No. D3125, Guangzhou Magen Biotechnology Co., Ltd., Guangzhou, China) according to the manufacturer’s protocol. After extracting the DNA, voucher specimens were retrieved and morphologically observed following Tixier et al. [19] to confirm molecular assignments. Three molecular markers (COI mtDNA, 12S rRNA, and ITS) were applied. The primers used to amplify these fragments were as follows: COI 5′-3′ GGTCAACAAATCATAAAGATATTGG and 3′-5′ TAAACTTCTGGATGTCCAAAAAATC [31,32]; 12S rRNA 5′-3′ TACTATGTTACGACTTAT and 3′-5′ TACCAGATTAAAATCTCAC [33]; and ITS 5′-3′ AGAGGAAGTAAAAGTCGTAACAAG and 3′-5′ ATATGCTTAAATTCAGGGGG [34]. For 12S rRNA and ITS, because a very low amount of DNA was obtained, an additional PCR was carried out, and the nested primers were designed based on the previous first PCR product, as follows: 12S rRNA 5′-3′ TGAACTAAATTAATTGGCGGCAA and 3′-5′ CTAAGGAGAGTGACGGGCAA; and ITS 5′-3′ GCGAATGGTGCGTGTATGAT and 3′-5′ CAGAAGTGTCCGTGCTGAAA. The PCRs were performed in a 20 μL volume, containing 1 μL of mite DNA, 10 μL of buffer 2 × 0.1 μL of each primer, and 8.8 μL of water. The thermal cycling conditions were for the COI marker: 95 °C for 3 min, followed by 45 cycles of 95 °C for 15 s, 45 °C for 15 s, 72 °C for 15 s, and 72 °C for 1 min; for the 12S rRNA marker: 95 °C for 3 min, followed by 45 cycles of 95 °C for 15 s, 42 °C for 15 s, 72 °C for 15 s, and 72 °C for 1 min, and 1 μL of PCR product was used for the second PCR: 95 °C for 3 min, followed by 45 cycles of 95 °C for 15 s, 52 °C for 15 s 72 °C for 15 s, and 72 °C for 1 min; and for the ITS marker: 95 °C for 3 min, followed by 45 cycles of 95 °C for 15 s, 50.5 °C for 15 s, 72 °C for 15 s, and 72 °C for 1 min, and 1 μL of product was used for the second PCR: 95 °C for 3 min, followed by 45 cycles of 95 °C for 15 s, 52 °C for 15 s and 72 °C for 15s, and 72 °C for 1 min.

Electrophoresis was carried out on 1.5% agarose gel in 1 × TBE buffer for 30 min at 120 volts. PCR products were sequenced using kits (Tsingke Biotechnology Co., Ltd., Beijing, China). The sequences were analyzed for both forward and reverse amplification fragments and aligned using ClustalW^®^.

Genetic distances (using the Kimura 2-parameter model) were calculated to compare DNA sequences. Maximum likelihood phylogenetic trees (using CART model) were built in MEGA 7.0, using the bootstrap method with 1000 replications to assess relationships. Molecular sequences of outgroup species *Amblyseius swirskii* Athias-Henriot were retrieved from the NCBI (National Center for Biotechnology Information) (Table 2). Morphological characteristics of all voucher specimens after DNA extracting were checked carefully.

## 3. Results

### 3.1. Morphological Analysis

Based on the specimens from Bajiaozhai National Forest Park, morphological characters of this putative *Euseius* sp. female are described as follows (Figure 1A–E and Figure 2A–C). While some distinguished morphological characteristics of putative *Euseius* sp., *E. nicholsi*, and *E. oolong* are listed in Table 1.

*Euseius* sp.

Females (n = 12) Dorsum (Figure 1A). Dorsal setal pattern 10A: 9B. Dorsal shield smooth, with anterior reticulation and lateral striation, 316 (299–335) long and 239 (223–254) wide, distances between setae j1–J5 307 (290–327) and s4–s4 183 (170–198), shield nearly oval, constricted at level of R1; r3 and R1 out of the dorsal shield, r3 at level of z4, R1 at level of Z1. All dorsal shield setae smooth and setiform, except Z5 slightly serrated. Seven pairs of solenostomes on dorsal shield (gd1, gd2, gd4, gd5, gd6, gd8, and gd9). Length of setae: j1 29 (25–33), j3 25 (22–31), j4 10 (8–11), j5 10 (8–12), j6 11 (9–14), J2 13 (10–15), J5 5 (4–6), z2 17 (15–22), z4 18 (16–24), z5 10 (7–12), Z1 12 (10–14), Z4 14 (12–17), Z5 59 (45–63), s4 28 (25–34), S2 18 (15–21), S4 24 (19–27), S5 26 (21–30), r3 14 (10–16), and R1 13 (10–15).

Venter (Figure 1B). Ventral setal pattern JV-3: ZV. All ventral setae smooth. Sternal shield lightly reticulated, posterior margin not obvious, 79 (75–84) wide with three pairs of setae st1 29 (24–32), st2 28 (23–32), and st3 27 (24–30), and two pairs of poroids (pst1–pst2), distance between st1–st3 54 (50–59), st2–st2 62 (58–65) and st3–st3 73 (68–77). Metasternal platelets drop-shaped, each with one metasternal seta, st4 25 (22–28) and a poroid (pst3). Genital shield smooth, 80 (76–85) wide, with one pair of thin genital setae st5 29 (27–32), distance between st5–st5 76 (70–86); one pair of associated poroids on soft cuticle near to the posterior corners of the shield. Ventrianal shield smooth, vase-shaped, anterior part strongly sclerotized with a hat-shaped structure, 93 (83–102) long, 47 (42–56) wide at level of ZV2, 75 (70–81) wide at level of anus, with three pairs of thin pre-anal setae JV1 29 (26–33), JV2 27 (25–32), and ZV2 21 (18–24); Pa 14 (12–17) and Pst 15 (13–17) long. Solenostomes gv3 crescentic, posteromesad JV2, distance between them 26 (24–29). Opisthogastric soft cuticle with four pairs of setae, ZV1 23 (20–29), ZV3 12 (9–14), JV4 12 (9–15), and JV5 30 (23–34) long. All ventral setae thin, except JV5, thick. Two pairs of metapodal platelets, primary ones 21 (16–24) long, 4 (3–5) wide, and secondary ones 9 (8–12) long, 1 (1–2) wide.

Peritreme. Peritreme extending to the level between setae j3 and z2.

Chelicera (Figure 1C). Fixed digit 23 (20–26) long, with four teeth and *pilus dentilis*; movable digit 22 (20–24) long, with one tooth.

Spermatheca (Figure 1D). Calyx tubular, with distal half lightly sclerotized, 11 (10–12) long, distal flaring, 3 (3–4) wide at the opening, atrium C-shaped, 2 (1–2) in depth, incorporated within the calyx; major duct without neck, narrow, and minor duct visible.

Legs. Genua formula for leg I 2-2/2, 1/1-2, leg II 2-2/2, 0/0-1, leg III 1-2/2, 1/0-1, and leg IV 1-2/2, 1/0-1. Tibia III with one macroseta, Sti III 24 (19–27). Genu, tibia, and basitarsus IV each with one macroseta (Figure 1E), Sge IV 41 (33–48), Sti IV 32 (27–37), and St IV 51 (43–62).

Males (n = 3) Dorsum (Figure 2A). Dorsal shield with anterior reticulation and lateral striation, 243 (242–243) long and 190 (167–205) wide at level of s6, shield nearly oval, 19 pairs of dorsal setae, all smooth, except Z5 slightly serrated. Seven pairs of solenostomes on dorsal shield (gd1, gd2, gd4, gd5, gd6, gd8, and gd9). Length of setae: j1 27 (26–28), j3 19 (18–22), j4 10 (9–11), j5 10 (9–13), j6 12 (11–13), J2 13 (12–13), J5 5 (4–5), z2 18 (16–22), z4 22 (18–25), z5 11 (10–14), Z1 13 (11–14), Z4 14 (12–16), Z5 48 (46–51), s4 30 (27–34), S2 19 (18–20), S4 22 (21–24), S5 22 (20–24), r3 14 (12–16), and R1 13 (12–14).

Venter (Figure 2B). Sternogenital shield 119 (114–126) long, reticulated. Wider 81 (78–84) between coxae II–III than at posterior corners 59 (58–59). Five pairs of sternogenital setae (st1–st5), st1 26 (24–30), st2 23 (22–24), st3 21 (19–23), st4 21 (20–22), and st5 22 (21–22), and three pairs of poroids (pst1–pst3). Ventrianal shield subtriangular, 92 (90–97) long, 154 (140–167) wide at level of anterior corners; transversally reticulated; reticulation on anterior part of shield more accentuated; with three pairs of thin pre-anal setae JV1 24 (22–26), JV2 21 (20–22), and ZV2 19 (17–21); Pa 13 (12–13) and Pst 12 (10–14) long. Solenostome gv3 crescentic, posteromedian to JV2, distance between solenostomes 23 (21–25). Opisthogastric soft cuticle with one pair of setae, JV5 25 (20–28) long.

Peritreme. Peritreme extending at level between setae j3 and z2. Peritrematal shield fused with dorsal shield.

Chelicera and spermatodactyl (Figure 2C). Cheliceral dentition not discernible in the examined specimens. Fixed digit 20 (19–20) long, movable digit 20 (18–22) long. Spermatodactyl L-shaped; shaft 25 (24–27), foot 7 (5–9) long.

Legs. Chaetotaxy of genua similar to female. Macroseta on genu I not differentiated. Macrosetae on tibia III 21 (17–24), genu IV 39 (38–40), tibia IV 30 (25–33), and tarsus IV 48 (42–53).

Morphological characteristics of *E. oolong* based on paratypes were re-drawn for comparison (Figure 3) and are summarized in Table 1.

### 3.2. Material Measured

Seven ♀♀, Bajiaozhai National Forest Park, Ziyuan County, Guilin City, Guangxi Province (slides no. GL-0591, GL-0592, GL-0841, GL-0561, GL-0562, GL-0951, and GL-0952); two ♀♀ (slides no. GL-0953 and GL-0601) on *Cyclobalanopsis glauca* (Fagaceae), 30 October 2020, Fang X.D. coll.; two ♀♀ (slides no. GL-0611 and GL-0612) on *Sabina squamata* (Cupressaceae), same locality, date, and collector; one ♀ (slide no. GL-0681) on *Smilax glaucochina* (Liliaceae), same locality, date, and collector; two ♂♂ (slides no. GL-0641 and GL-0671) and one ♂ (slide no. GL-0672 ) on *Bambusa* sp. (Poaceae), same locality, date, and collector. 

### 3.3. Molecular Analysis

Genetic distances between the putative *Euseius* sp. and two morphologically similar species (*E. nicholsi* and *E. oolong*) ranged between 0 and 4% for COI mtDNA, between 0 and 2% for 12S rRNA, and were 0 for ITS. These values were markedly lower than those observed between *Euseius* sp. specimens and *E. finlandicus* (23–24% for COI mtDNA, 21–22% for 12S rRNA, and 2% for ITS) or *A. swirskii* (30–33% for COI mtDNA, 34–39% for 12S rRNA, and 18% for ITS) (Table 3, Table 4, and Table 5).

In the phylogenetic tree based on COI sequences (Figure 1A), the putative *Euseius* sp., *E. nicholsi*, and *E. oolong* clustered together within a single clade, supported by bootstrap values of 92–98%, and were clearly separated from the outgroup *A. swirskii*. A similar pattern was recovered with the 12S rRNA marker (Figure 1B), where all specimens of *E. nicholsi*, *E. oolong*, and the putative *Euseius* sp. formed a single well-supported lineage (bootstrap > 90%). The ITS tree (Figure 1C) showed identical clustering, with no divergence among the three taxa. These consistent topologies indicated that the three groups represent a single evolutionary lineage and provided further support for conspecificity.

### 3.4. Comparative Synthesis of Morphological and Molecular Results

When the three taxa were compared directly, the specimens from Bajiaozhai (putative *Euseius* sp.) were generally smaller in body size than *E. nicholsi* but overlapped partly with *E. oolong*. Variation was also observed in the number and shape of metapodal platelets: the putative *Euseius* sp. showed one or two pairs, either hooked or unhooked, whereas *E. nicholsi* typically has a single hooked pair and *E. oolong* usually two unhooked pairs. The spermathecal calyx of the Bajiaozhai specimens was tubular and only lightly sclerotized distally, in contrast to the funnel-shaped calyx of *E. nicholsi* and the cup-shaped calyx of *E. oolong*. In addition, the dorsal shield bore seven solenostomes, the same as in *E. nicholsi* but more than the five solenostomes generally described for *E. oolong*. Measurements of setae (e.g., j3, Sge IV, Sti IV, and St IV) were intermediate between the two named species (Table 6).

The molecular markers confirmed these observations. Genetic distances among the three taxa were minimal (COI: 0–4%; 12S rRNA: 0–2%; and ITS: 0%), far below interspecific levels reported for other phytoseiids. In all phylogenetic reconstructions (Figure 4A–C), the Bajiaozhai specimens, *E. nicholsi*, and *E. oolong* formed a single clade, clearly distinct from the reference species *E. finlandicus* and the outgroup *A. swirskii*. These results showed that the morphological differences were intraspecific rather than interspecific, and that *E. oolong* was best treated as a junior synonym of *E. nicholsi*.

**Table 3 insects-16-00950-t003:** Genetic distances between cytochrome oxidase subunit 1 (COI) sequences of putative *Euseius* sp., *E. nicholsi*, and *E. oolong* using the Kimura 2-parameter model.

	Putative *Euseius* sp. (OQ813513)	Putative *Euseius* sp. (OQ813514)	Putative *Euseius* sp. (OQ813515)	*E. oolong* (OQ809342)	*E. oolong* (OQ809343)	*E. oolong* (OQ809344)	*E. nicholsi* (OR610851)	*E. nicholsi* (OR610849)	*E. nicholsi* (OR610850)	*E. finlandicus* (OL863178)	*Amblyseius* *swirskii* (MW074356)
Putative *Euseius* sp. (OQ813513)											
Putative *Euseius* sp. (OQ813514)	0.00										
Putative *Euseius* sp. (OQ813515)	0.01	0.01									
*E. oolong* (OQ809342)	0.02	0.02	0.02								
*E. oolong* (OQ809343)	0.02	0.02	0.02	0.00							
*E. oolong* (OQ809344)	0.02	0.02	0.02	0.00	0.00						
*E. nicholsi* (OR610851)	0.02	0.02	0.02	0.00	0.00	0.00					
*E. nicholsi* (OR610849)	0.01	0.01	0.01	0.02	0.02	0.02	0.02				
*E. nicholsi* (OR610850)	0.04	0.04	0.04	0.04	0.04	0.04	0.04	0.04			
*E. finlandicus* (OL863178)	0.23	0.23	0.23	0.23	0.23	0.23	0.23	0.24	0.24		
*Amblyseius swirskii* (MW074356)	0.31	0.31	0.31	0.30	0.30	0.30	0.30	0.31	0.33	0.30	

**Table 4 insects-16-00950-t004:** Genetic distances between 12S rRNA sequences of putative *Euseius* sp., *E. nicholsi*, and *E. oolong* using the Kimura 2-parameter model.

	Putative *Euseius* sp. (OQ799396)	Putative *Euseius* sp. (OQ799397)	Putative *Euseius* sp. (OQ799398)	*E. oolong* (OQ780361)	*E. oolong* (OQ780362)	*E. oolong* (OQ780363)	*E. nicholsi* (PRJNA1022847)	*E. nicholsi* (OR614377)	*E. nicholsi* (OR614378)	*E. finlandicus* (FJ404578)	*Amblyseius* *swirskii* (KX064694)
Putative *Euseius* sp. (OQ799396)											
Putative *Euseius* sp. (OQ799397)	0.00										
Putative *Euseius* sp. (OQ799398)	0.01	0.01									
*E. oolong* (OQ780361)	0.02	0.02	0.01								
*E. oolong* (OQ780362)	0.02	0.02	0.01	0.02							
*E. oolong* (OQ780363)	0.02	0.02	0.01	0.02	0.00						
*E. nicholsi* (PRJNA1022847)	0.02	0.02	0.02	0.02	0.00	0.00					
*E. nicholsi* (OR614377)	0.01	0.01	0.00	0.01	0.01	0.01	0.01				
*E. nicholsi* (OR614378)	0.02	0.02	0.01	0.02	0.00	0.00	0.00	0.01			
*E. finlandicus* (FJ404578)	0.21	0.21	0.21	0.21	0.22	0.22	0.22	0.21	0.22		
*Amblyseius swirskii* (KX064694)	0.34	0.34	0.34	0.34	0.35	0.35	0.36	0.34	0.35	0.39	

**Table 5 insects-16-00950-t005:** Genetic distances between ITS sequences of putative *Euseius* sp., *E. nicholsi*, and *E. oolong* using the Kimura 2-parameter model.

	Putative *Euseius* sp. (OQ799393)	Putative *Euseius* sp. (OQ799394)	Putative *Euseius* sp. (OQ799395)	*E. oolong* (OQ780935)	*E. oolong* (OQ780936)	*E. oolong* (OQ780937)	*E. nicholsi* (OR607938)	*E. nicholsi* (OR607939)	*E. nicholsi* (OR607948)	*E. finlandicus* (MT436747)	*Amblyseius* *swirskii* (MT436721)
Putative *Euseius* sp. (OQ799393)											
Putative *Euseius* sp. (OQ799394)	0.00										
Putative *Euseius* sp. (OQ799395)	0.00	0.00									
*E. oolong* (OQ780935)	0.00	0.00	0.00								
*E. oolong* (OQ780936)	0.00	0.00	0.00	0.00							
*E. oolong* (OQ780937)	0.00	0.00	0.00	0.00	0.00						
*E. nicholsi* (OR607938)	0.00	0.00	0.00	0.00	0.00	0.00					
*E. nicholsi* (OR607939)	0.00	0.00	0.00	0.00	0.00	0.00	0.00				
*E. nicholsi* (OR607948)	0.00	0.00	0.00	0.00	0.00	0.00	0.00	0.00			
*E. finlandicus* (MT436747)	0.02	0.02	0.02	0.02	0.02	0.02	0.02	0.02	0.02		
*Amblyseius swirskii* ( MT436721)	0.18	0.18	0.18	0.18	0.18	0.18	0.18	0.18	0.18	0.18	

**Table 6 insects-16-00950-t006:** Key diagnostic traits comparing putative *Euseius* sp., *E. nicholsi*, and *E. oolong*.

Character	Putative *Euseius* sp.	*E. nicholsi*	*E. oolong*
Female dorsal shield length (µm)	299–335	352–381	323–361
Dorsal solenostomes	7	7	5 (sometimes 7, weak gd1)
Spermathecal calyx	Tubular, weakly sclerotized	Funnel-shaped, flaring	Cup-shaped
Metapodal platelets	1–2 pairs, hooked/unhooked	1 pair, mostly hooked	2 pairs, unhooked
Fixed digit teeth	4	3–4	5
j3 seta (µm)	22–31	27–33	20–25
St IV (µm)	43–62	56–73	47–61
Molecular divergence	COI 0–4%; 12S 0–2%; ITS 0%	Same	Same
Phylogenetic placement	Same clade	Same clade	Same clade

## 4. Discussion

The comparative analysis presented in the Results section showed that morphological variation among the Bajiaozhai specimens, *E. nicholsi*, and *E. oolong* was minor and largely overlapped. Features such as differences in spermathecal calyx shape, dorsal solenostomes, metapodal platelets, and setal lengths fell within the range of variation observed for other phytoseiid mites. To determine whether these differences were taxonomically significant, molecular evidence was compared with previous studies of genetic divergence in Phytoseiidae.

For the COI marker, interspecific divergence within phytoseiid genera such as *Kampimodromus* and *Typhlodromus* can reach 17–18%, whereas intraspecific variation is typically very low, for example, 0–0.4% in *Neoseiulus californicus*, 4–6% in *K. aberrans*, and 6% in *K. hmiminai* [16,18]. By contrast, the specimens examined here showed only 0–4% divergence, indicating they belonged to the same species. For the 12S rRNA fragment, the maximum divergence between closely related species has been reported at 10% [18], while intraspecific variation generally ranges from 0 to 3% [19,24]. The 0–2% divergence recorded in our study fell within this intraspecific range. Similarly, ITS divergence among congeneric phytoseiids often ranges from 3 to 7% [18], but in our data, the divergence was 0%. These values demonstrated that the putative *Euseius* sp., *E. nicholsi*, and *E. oolong* were conspecific. The congruent clustering of all three taxa in COI, 12S rRNA, and ITS phylogenies (Figure 4A–C), each with high bootstrap support, provided strong molecular evidence that *E. nicholsi* and *E. oolong* cannot be separated as distinct species.

Both mitochondrial (COI, 12S) and nuclear (ITS) markers therefore supported synonymy. The phylogenetic trees (Figure 4A–C) clustered all three taxa into a single clade with low genetic distances, clearly distinct from *E. finlandicus* and *A. swirskii*. This pattern was consistent with earlier studies, where low bootstrap values and minor genetic variation reflected population-level differentiation rather than species boundaries [16,17,35].

Taken together, the molecular results confirmed that the morphological differences observed among the three taxa represented intraspecific variation. Consequently, *E. oolong* should be regarded as a junior synonym of *E. nicholsi*, and the Bajiaozhai specimens can be confidently identified as *E. nicholsi*. This conclusion contributes to a clearer understanding of phytoseiid systematics and highlights the importance of integrative taxonomy for resolving synonymy issues that complicate biodiversity assessment and biological control programs.

## 5. Conclusions

According to Tixier et al. [6], the minimal difference of setae length between the means of two specimen lots belonging to two species should be 10.58 μm (for setae < 65 μm) and 33.99 μm (for setae > 65 μm).

Therefore, the two following points can be concluded:The specimens of *Euseius* sp. from Bajiaozhai National Forest Park were morphologically characterized as *E. nicholsi*, due to their high similarity. Based on molecular analyses, the specimens identified as *E. nicholsi* exhibited minimal genetic divergence from *Euseius* sp., indicating they likely represented the same species. Therefore, the morphological differences between *Euseius* sp. and *E. nicholsi* were best interpreted as intraspecific variation.Molecular evidence indicated that *E. nicholsi* and *E. oolong* are conspecific, with *E. oolong* treated as a synonym of *E. nicholsi*. Variations in body size, metapodal platelets, digit dentition, and setal lengths reflect intraspecific diversity.

According to Ehara and Lee [9], *E. nicholsi* is characterized by a pair of hook-shaped metapodal platelets, which appear to originate from the joining of two metapodal platelets. However, our molecular analysis revealed highly similar sequences among specimens with both hooked and non-hooked metapodal platelets, as well as among those bearing one or two pairs of platelets (Table 1). Additionally, no molecular divergence was detected among specimens exhibiting slight differences in the number of teeth on the fixed digit. These findings suggest that intraspecific variation is more extensive than previously anticipated.

## Figures and Tables

**Figure 1 insects-16-00950-f001:**
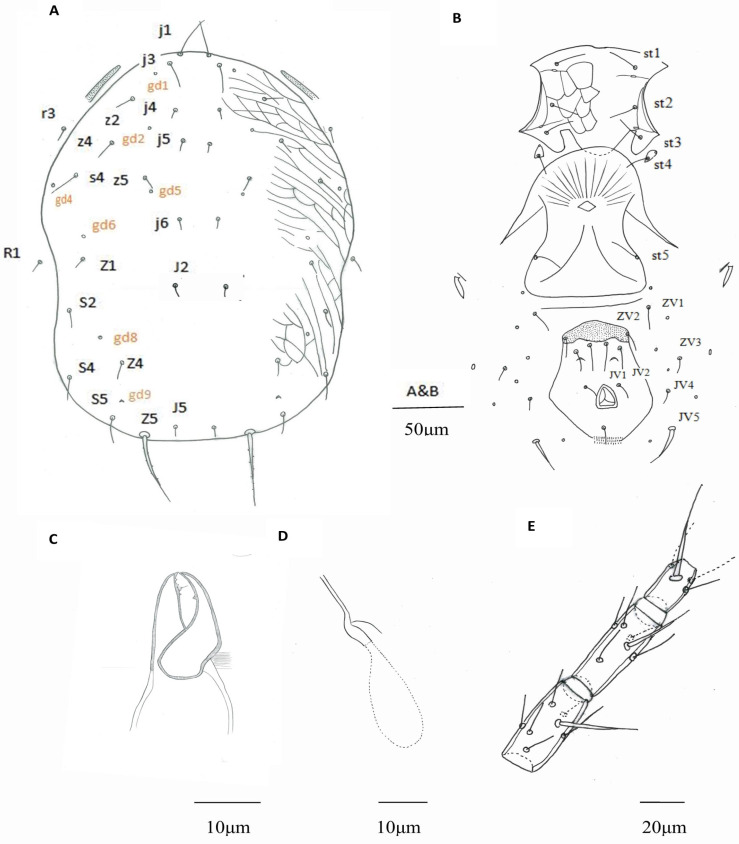
*Euseius* sp., female. (**A**) Dorsal shield; (**B**) Ventral idiosoma; (**C**) Chelicera; (**D**) Spermatheca; (**E**) Leg IV, genu-basitarsus.

**Figure 2 insects-16-00950-f002:**
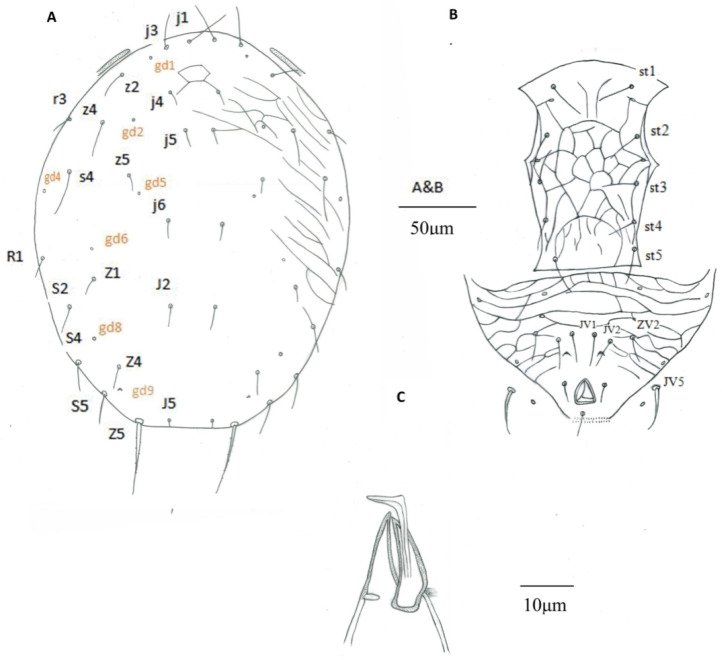
*Euseius* sp., male. (**A**) Dorsal idiosoma; (B) Ventral idiosoma; (C) Spermatodactyl.

**Figure 3 insects-16-00950-f003:**
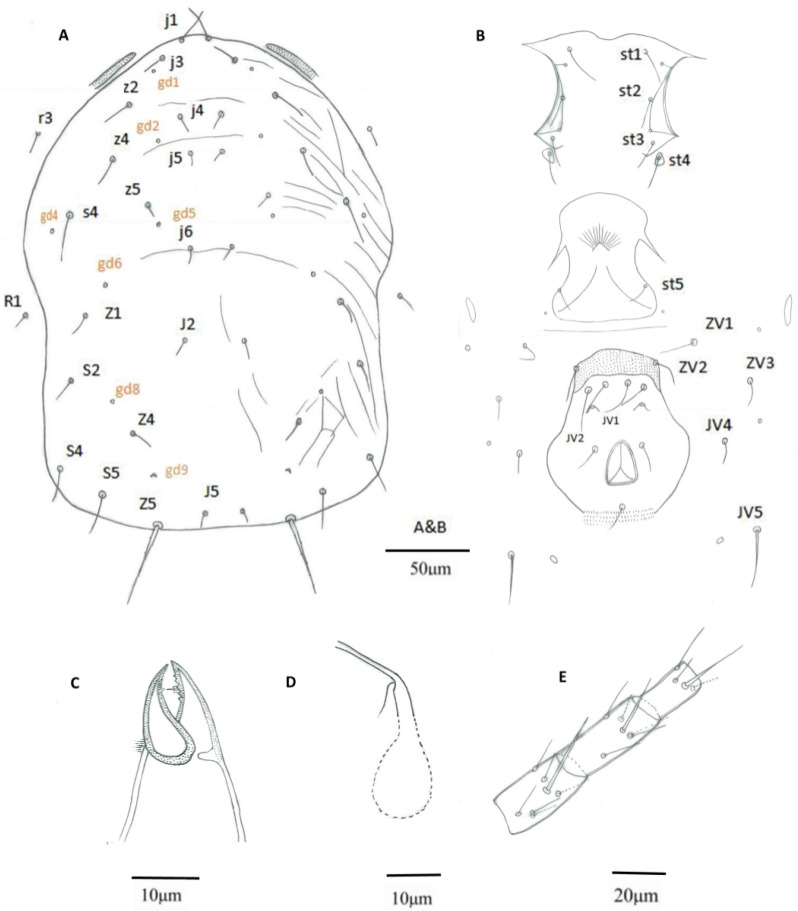
(From paratypes). *E. oolong* Female, (**A**) Dorsal shield; (**B**) Ventral idiosoma; (**C**) Chelicera; (**D**) Spermatheca; (**E**) Leg IV, genu-basitarsus.

**Figure 4 insects-16-00950-f004:**
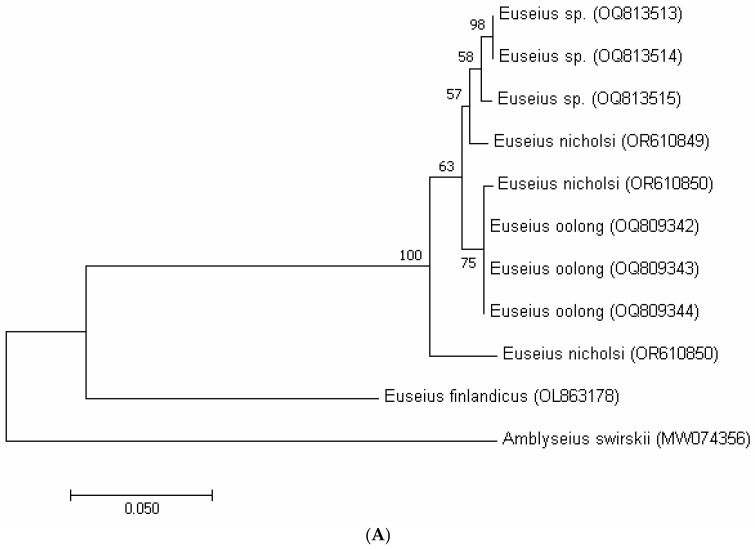
Maximum likelihood phylogenetic trees including putative *Euseius* sp. from Bajiaozhai National Forest Park, *E. oolong* from ZIGAS, *E. nicholsi* from ZIGAS, and also Shaoguan newly collected specimens (*Amblyseius swirskii* as the outgroup, *E. finlandicus* as the reference) obtained with (**A**)—COI mtDNA, (**B**)—12S rRNA and (**C**)—ITS markers.

**Table 1 insects-16-00950-t001:** Morphological characteristics and measurements differentiating putative *Euseius* sp., *E. nicholsi*, and *E. oolong*.

	Putative *Euseius* sp. ^a^	*E. nicholsi* ^b^	*E. nicholsi* ^c^	*E. nicholsi* ^d^	*E. oolong* ^e^	*E. oolong* ^f^
Dorsal shield length (µm)	316 (299–335)	360	365–381	352 (348–357)	341 (324–361)	323 (319–328)
Dorsal shield width (µm) (s4 level)	239 (223–254)	260	265–282	251 (246–257)	246 (236–266)	238 (233–243)
Dorsal shield	anterior reticulation and lateral striation	most surface reticulated	most surface reticulated	most surface reticulated	laterally reticulated	laterally reticulated
Solenostome on dorsal shield	*gd1*, *gd2*, *gd4*, *gd5*, *gd6*, *gd8*, *gd9*	*gd1*, *gd2*, *gd4*, *gd5*, *gd6*, *gd8*, *gd9*		*gd1*, *gd2*, *gd4*, *gd5*, *gd6*, gd8, gd9	*gd2*, *gd5*, *gd6, gd8, gd9*	*gd1*, *gd2*, *gd4*, *gd5*, *gd6*, *gd8*, *gd9*
Calyx of spermatheca	tubular, with distal half only lightly sclerotized; atrium c-shaped incorporated within the calyx	tubular, with basal half swollen	funnel-shaped, distal flaring; atrium knobbed, connected directly with calyx	funnel-shaped, distal flaring; atrium knobbed, connected directly with calyx	cup-shaped with distal half only lightly sclerotized; atrium c-shaped incorporated within the calyx	cup-shaped with distal half only lightly sclerotized; atrium c-shaped incorporated within the calyx
Metapodal platelet	one or two pairs, hooked or not	one pair, hooked	one pair, hooked or not	one pair, hooked or not	two pairs, un-hooked	one pair, hooked
Number of teeth on the fixed digit of the chelicerae	4		3–4	4	5	5
Seate j3	25 (22–31)	27 (26–28)	30	31 (30–33)	22 (20–25)	22 (20–25)
Macroseta Sge IV	41 (33–48)	50 (48–53)	52	48 (45–51)	44 (37–50)	50 (47–54)
Macroseta Sti IV	32 (27–37)	38 (38–39)	41	34 (33–35)	34 (27–37)	33 (32–34)
Macroseta St IV	51 (43–62)	68 (60–73)	66	56 (55–57)	58 (50–61)	50 (47–51)

^a^ From 12 female specimens in Bajiaozhai National Forest Park; ^b^ From Holotype and reference [10]; ^c^ From reference [1]; ^d^ From newly collected specimens; ^e^ From reference [10]; ^f^ From two paratypes deposited at IZGAS and also newly collected specimens.

**Table 2 insects-16-00950-t002:** Characteristics of *E. nicholsi*, *E. finlandicus*, and outgroup species *Amblyseius swirskii*, and GenBank accession numbers of Cytochrome c Oxidase subunit I (COI), 12S rRNA, and internal transcribed spacer (ITS) markers.

Species	Country	Locality	Plant Supports	GenBank Accession Number of COI Sequences	GenBank Accession Number of 12S rRNA Sequences	GenBank Accession Number of ITS Sequences
Putative *Euseius* sp.	China	Guilin	*Cyclobalanopsis glauca*	OQ813513 OQ813514 OQ813515	OQ799396 OQ799397 OQ799398	OQ799393 OQ799394 OQ799395
*E. oolong*	China	Guangzhou	*Bauhinia purpurea*	OQ809342 OQ809343 OQ809344	OQ780361 OQ780362 OQ780363	OQ780935 OQ780936 OQ780937
*E. nicholsi*	China	Guangzhou Shaoguan Shaoguan	*Bauhinia purpurea* *Eurya macar* *tneyi* *Ficus hispida*	OR610851 OR610849 OR610850	SRR26247364 OR614377 OR614378	OR607938 OR607939 OR607948
*E. finlandicus*	Russia France Turkey	Sochi Unknown Unknown	*Ulmus* sp. *Prunus cerasus* Unknown	OL863178	FJ404578	MT436747
*Amblyseius swirskii*	France The Netherlands Turkey	Unknown Unknown Unknown	Unknown Rearing population Unknown	MW074356	KX064694	MT436721

## Data Availability

The data sheets are available and can be requested from the corresponding author upon reasonable request.

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
