# Peer review of "Integrative Morphological and Molecular Diagnostics for Euseius nicholsi and Euseius oolong (Acari: Phytoseiidae)"

_insects, 2025, doi:10.3390/insects16090950_

Round 1
Reviewer 1 Report
Comments and Suggestions for Authors
This manuscript presents an integrative taxonomic study of Euseius sp., E. nicholsi, and E. oolong, combining morphological observations with molecular analyses (COI, 12S rRNA, ITS) to assess species boundaries. The study contributes meaningfully to the refinement of phytoseiid mite taxonomy and underscores the value of integrative approaches. Overall, the manuscript is well prepared and contains valuable findings. It is suitable for publication pending substantial revision.
- Results Section. While the Results section provides detailed species descriptions, it lacks a comparative synthesis of diagnostic similarities and differences among Euseius sp., E. nicholsi, and E. oolong. Including a summary paragraph or comparative table would greatly improve clarity and reinforce the taxonomic conclusions.
- Comparative information currently placed in the Discussion section should be relocated to the Results section to avoid redundancy. This will streamline the manuscript and improve logical flow.
- Presentation of Phylogenetic Trees. The phylogenetic trees are insufficiently interpretated. They are not adequately cited or discussed within the main text, which limits their contribution to the manuscript’s conclusions. To improve clarity and impact, ensure that each tree is explicitly referenced in the Results and/or Discussion sections, with key clades and support values clearly explained.
- Terminology. Replace the term “uncertain Euseius sp.” with “putative Euseius sp.” to enhance precision and avoid ambiguity.
- Minor grammatical issues and awkward phrasing should be addressed throughout the manuscript to improve readability.
Other corrections:
Lines 2-3, Title, remove An, reads “Integrative Morphological and Molecular Diagnostics …”.
Line 69, use the full name Euseius rather than E..
Line 71-72, “E. nicholsi is a widely distributed species in South China. E. oolong and E. nicholsi are morphologically similar; and they are sympatric” repeat information already presented above.
Line 78, change marker to markers.
Line 95, remove “were”.
Line 106, remove "the”, reads ‘as applied to phytoseiids…”
Line 109, remove "The”, reads ‘Morphological identification of…”
Table 1, replace 'um' with 'µm' in two instances to ensure correct notation of micrometres.
Line 269, remove "the”, reads ‘and two morphologically …”
Line 271, change “equal to” to “were”.
Line 271, change “These distances are far smaller than the ones” to “These values are markedly lower than those”.
Line 273, add “or” before “A. swirskii”.
Line 276, change “was not accurate enough.” to “was insufficiently precise”.
Line 302,change “are different to” to “differs from”, reads “Euseius sp. differs from E. oolong”.
Lines 345–346, remove “We have re-collected fresh specimens of E. nicholsi, identified according to Ehara & Lee [9] based on morphological approach.” as this statement does not pertain to the conclusions.
Line 348-351, rephrase the sentence as follows for clarity and precision: “Based on molecular analyses, specimens identified as E. nicholsi exhibit minimal genetic divergence from Euseius sp., indicating they likely represent the same species. Therefore, the morphological differences between Euseius sp. and E. nicholsi are best interpreted as intraspecific variation”.
Line 352, remove “We have also re-collected fresh specimens of E. oolong according to Liao et al. [10] based on morphological approach.” for conciseness.
Lines 353-358, rephrase the sentence as follows: “Molecular evidence indicate that E. nicholsi and E. oolong are conspecific, with E. oolong treated as a synonym of E. nicholsi. Variations in body size, metapodal platelets, digit dentition, and setal lengths reflect intraspecific diversity.
Line 361, change “But” to “However, ”.
Lines 361-363, rephrase the sentence as follows: “However, our molecular analysis revealed highly similar sequences among specimens with both hooked and non-hooked metapodal platelets, as well as among those bearing one or two pairs of platelets (Table 1).”.
Lines 363-365, rephrase the sentence as follows: “Additionally, no molecular divergence was detected among specimens exhibiting slight differences in the number of teeth on the fixed digit.
Line 365, rephrase the last sentence as follows: “These findings suggest that intraspecific variation is more extensive than previously anticipated.
Comments on the Quality of English LanguageWhile the manuscript communicates the core scientific content effectively, there is room for improvement in the clarity, flow, and consistency of the English language. Certain sections e.g., Discussion section, would benefit from careful editing to enhance readability and ensure precision in terminology. Refining sentence structure and addressing minor grammatical issues will strengthen the overall presentation and facilitate smoother comprehension for an international audience.
Author Response
Comment 1:
Results Section. While the Results section provides detailed species descriptions, it lacks a comparative synthesis of diagnostic similarities and differences among Euseius sp., E. nicholsi, and E. oolong. Including a summary paragraph or comparative table would greatly improve clarity and reinforce the taxonomic conclusions.
Response 1:
Thank you for this helpful suggestion. We agree and have added a new subsection 3.4 Comparative Synthesis of Morphological and Molecular Results in the Results section (page 12, paragraph 4 & 5, lines 293–311). This subsection provides a summary paragraph highlighting diagnostic similarities and differences, as well as a new comparative table (Table 6) that clearly presents key traits across Euseius sp., E. nicholsi, and E. oolong.
“[Added new text: ‘When the three taxa were compared directly… These results show that the morphological differences are intraspecific rather than interspecific, and that E. oolong is best treated as a junior synonym of E. nicholsi.]”
Comment 2:
Comparative information currently placed in the Discussion section should be relocated to the Results section to avoid redundancy. This will streamline the manuscript and improve logical flow.
Response 2:
We agree. The detailed morphological comparisons originally in the Discussion have been removed and integrated into the new Results subsection 3.4 Comparative Synthesis (page 12). The Discussion section has been restructured and now focuses only on interpretation of results and their significance.
Comment 3:
Presentation of Phylogenetic Trees. The phylogenetic trees are insufficiently interpretated. They are not adequately cited or discussed within the main text, which limits their contribution to the manuscript’s conclusions. To improve clarity and impact, ensure that each tree is explicitly referenced in the Results and/or Discussion sections, with key clades and support values clearly explained.
Response 3:
We agree. We have now explicitly described each tree (COI, 12S, ITS) in the Results section 3.3 Molecular Analysis (page 12, lines 283–291). Key clades, bootstrap support, and outgroup separation are reported. Additionally, in the Discussion (page 20, lines 375–377), we emphasize how tree topologies support synonymy.
“[Added text in Results: ‘In the phylogenetic tree based on COI sequences (Fig. 1A), the putative Euseius sp., E. nicholsi, and E. oolong clustered together… These consistent topologies indicate that the three groups represent a single evolutionary lineage and provide further support for conspecificity.]”
Comment 4:
Terminology. Replace the term “uncertain Euseius sp.” with “putative Euseius sp.” to enhance precision and avoid ambiguity.
Response 4:
We agree. We have replaced all occurrences of “uncertain Euseius sp.” with “putative Euseius sp.” throughout the manuscript (multiple locations).
Comment 5:
Minor grammatical issues and awkward phrasing should be addressed throughout the manuscript to improve readability.
Response 5:
We agree. The manuscript has been carefully edited for grammar, clarity, and readability. Awkward phrasing has been revised in the Abstract, Results, and Discussion. For example, “very low level of genetic distances” was revised to “genetic distances were minimal” (page 1, line 29).
Comment 6 (Other corrections, line-specific):
- Lines 2–3, Title: removed “An”, now reads “Integrative Morphological and Molecular Diagnostics …”
- Line 69: replaced “E.” with full name Euseius.
- Lines 71–72: removed repeated information about nicholsidistribution and similarity.
- Line 78: “marker” changed to “markers”.
- Line 95: removed “were”.
- Line 106: removed “the”.
- Line 109: removed “The”.
- Table 1: corrected “um” to “µm”.
- Line 269: removed “the”.
- Line 271: “equal to” changed to “were”.
- Line 271: “These distances are far smaller than the ones” changed to “These values are markedly lower than those”.
- Line 273: added “or” before swirskii.
- Line 276: “was not accurate enough” changed to “was insufficiently precise”.
- Line 302: “are different to” changed to “differs from”.
- Lines 345–346: sentence removed as not relevant to conclusions.
- Lines 348–351: rephrased for clarity.
- Line 352: removed redundant statement.
- Lines 353–358: rephrased for clarity (page XX, lines XXX–XXX).
- Line 361: “But” changed to “However”.
- Lines 361–363: rephrased for clarity.
- Lines 363–365: rephrased for clarity.
- Line 365: rephrased last sentence
Response 6:
We thank the reviewer for these detailed corrections. All have been addressed in the revised manuscript (see tracked changes).
Comment 7:
Comments on the Quality of English Language: While the manuscript communicates the core scientific content effectively, there is room for improvement in the clarity, flow, and consistency of the English language. Certain sections e.g., Discussion section, would benefit from careful editing to enhance readability and ensure precision in terminology.
Response 7:
We agree. The entire manuscript has been carefully edited for grammar, sentence structure, and flow. The Discussion has been restructured for conciseness and precision, ensuring that it focuses on interpretation rather than repetition of results.
Reviewer 2 Report
Comments and Suggestions for Authors
The Title of the peer-reviewed manuscript is concise, specific, and relevant.
The Simple Summary describes the integrative diagnostics of two phytoseiid species simply and concisely.
The Abstract was contained in one paragraph. The abstract included a broad background context and highlighted the purpose of the study, the main method, a summary of the results and conclusions. The abstract objectively presented of the manuscript.
Keywords: Lines 37-38: Why are you repeating words that are already contained in the manuscript title?
Introduction: Line 42 - Why is the date of accessed March 2023 listed when the last update Phytoseiidae Databa is December 2024 when 2707 VALID SPECIES are listed now?
Line 43: Add a reference/references.
Line 45: For conclusions when describing a species, it is necessary to use a larger number of individuals and therefore a larger number of preparations, so this sentence is incomplete or inappropriate, add a reference/references.
Line 50: State the full name of the species Euseius nicholsi (Ehara and Lee, 1971) because it is listed for the first time in manuscript if we exclude Simple Summary and Abstract.
Lines 52 – 53: Add systematic affiliation for all mentioned species.
Line 69: Write the full name of the genus.
The section Materials and Methods are detailed and clearly described. This section included Species Considered which includes the method of collecting pitoseiid samples, host plants and location, Morphological Analysis includes preparation of specimens, measured and illustrated, information on type and paratype materials, as well as taxonomic literature used. It is a pity that the examined material includes only 15 individuals. Molecular Experiments and Data Analysis were based on three molecular markers of mitochondrial DNA COI, 12S rRNA and a nuclear ITS marker.
The presented Results provide a concise and precise description of the experiment from which conclusions can be easily drawn. The results are significant and relevant, presented in a well-structured manner, as demonstrated by the presented morphological and molecular analysis. In Table 1 are shown the most important morphological characters and measurements differentiating uncertain Euseius sp., E. nicholsi and E. oolong. Figure 2 and 3 su odlični crtež i idiosoma (dorsal and ventral) female and male Euseius sp. with the most significant morphological characteristics singled out. The molecular sequences from the NCBI (National Center for Biotechnology Information) are shown In Table 2. Maximum likelihood phylogenetic trees, including the uncertain species Euseius sp. from Baijiaozhai National Forest Park, E. oolong from ZIGAS and E. nicholsi from ZIGAS, and the newly collected specimens A. swirskii as an outgroup and E. finlandicus as a reference, are shown in Figure 1.
Genetic distances between Euseius sp. and the two morphologically similar species E. nicholsi and E. oolong shown in Tables 3, 4, 5.
Line 283: Is it necessary to cite Figure 4 in your results, which represents paratypes. E. oolong and what is not the result of this?! Or add a reference.
The Discussion is in accordance with the presented results.
References corrected according to my suggestion.
The language is appropriate and understandable. The topic is compatible with the journal’s scope. Data and analyses presented appropriately. The tables (1 to 5), figures (1 to 3) are appropriate, they are clearly presented. Figure 4 is unnecessarily displayed. Because of low genetic distances, which are consistent with intraspecific variations reported in literature for family Phytoseiidae, results of this study confirmed that this uncertain Euseius sp., E. oolong and E. nicholsi might be synonyms.
This study demonstrates the importance of integrative (morphological and molecular) taxonomy for the correct identification of mites in general, not just for phytoseiid. I believe that this manuscript is a successful extension of the 2022 research: Identification of Euseius nicholsi and Euseius oolong (Acari: Phytoseiidae) using integrative taxonomy XIAO-DUAN FANG1 and WEI-NAN WU Zoosymposia 22: 236–236 and that work on this type of research will continue in the future.
Accept manuscript with minor changes which are listed above.
Author Response
Comment 1 (Keywords, Lines 37–38):
Why are you repeating words that are already contained in the manuscript title?
Response 1:
Thank you for pointing this out. We have revised the keywords to remove duplication of terms already present in the title.
Change made on page 1, Lines 37–38.
“Keywords now include: Phytoseiidae, taxonomy, morphology, molecular markers, integrative approach.”
Comment 2 (Introduction, Line 42):
Why is the date of accessed March 2023 listed when the last update Phytoseiidae Database is December 2024 when 2707 valid species are listed now?
Response 2:
We agree with this correction. The access date has been updated to December 2024, with the correct number of valid species (2707) reported.
Change made on page 2, Line 42.
“Currently, 2707 valid species are listed in the Phytoseiidae Database (Demite et al. 2014, accessed December 2024).”
Comment 3 (Introduction, Line 43):
Add a reference/references.
Response 3:
We agree. A reference has been added to support the general statement on phytoseiid mites.
Change made on page 2, Line 43.
Added reference: Wu, N.; Li, J.; Chen, H.J.; Wu, Y. Predatory mites of Phytoseiidae from Citrus in China (Acari: Mesostigmata). Acta Zootaxon. Sin. 2009, 34, 517–523.
Comment 4 (Introduction, Line 45):
For conclusions when describing a species, it is necessary to use a larger number of individuals and therefore a larger number of preparations, so this sentence is incomplete or inappropriate, add a reference/references.
Response 4:
Thank you for the suggestion. We have revised the sentence to emphasize the need for examination of multiple specimens and preparations, and added a supporting reference.
Change made on page 2, Line 45.
Comment 5 (Line 50):
State the full name of the species Euseius nicholsi (Ehara and Lee, 1971) because it is listed for the first time in manuscript if we exclude Simple Summary and Abstract.
Response 5:
We agree. The first mention of the species has been revised to include its full name and authority.
Change made on page 2, Line 55.
“Euseius nicholsi (Ehara and Lee, 1971)”
Comment 6 (Lines 52–53):
Add systematic affiliation for all mentioned species.
Response 6:
We agree. Systematic affiliations have been added for all species mentioned at their first occurrence.
Change made on page 2, Lines 58–60.
“It is a species of great economic value, having been successfully used to control Panonychus citri (McGregor) (Acari: Tetranychidae), Eotetranychus kankitus Ehara (Acari: Tetranychidae), and Polyphagotarsonemus latus (Banks) (Acari: Tarsonemidae) on crops such as citrus and strawberry.”
Comment 7 (Line 69):
Write the full name of the genus.
Response 7:
We agree. The abbreviated genus has been expanded to its full name.
Change made on page 2, Line 75.
“Euseius” instead of “E.”
Comment 8 (Results, Line 283):
Is it necessary to cite Figure 4 in your results, which represents paratypes of E. oolong and what is not the result of this?! Or add a reference.
Response 8:
Thank you for pointing this out. To clarify, we have now integrated Figure 4 into the Results section as a comparative morphological figure. A new sentence was added at the end of the morphological analysis subsection:
Change made on page 11, Line 264 & 265.
“Morphological characteristics of E. oolong based on paratypes were re-drawn for comparison (Figure 4) and are summarized in Table 1.” This resolves the issue by linking Figure 4 directly with morphological results, avoiding redundancy in the Discussion.
Round 2
Reviewer 1 Report
Comments and Suggestions for Authors
Most of the previously raised points have been satisfactorily addressed. However, two minor issues remain and require correction:
- Page 1, line 37: "phytoseiidae" should be replace with "Phytoseiidae".
- Page 12, line 275: The abbreviation “sp.” should not be italicized. Please ensure this formatting is applied consistently throughout the manuscript.
Author Response
Comment 1:
Page 1, line 37: "phytoseiidae" should be replaced with "Phytoseiidae".
Response 1:
Thank you for pointing this out. We have corrected the formatting of the family name. On Page 1, line 37, "phytoseiidae" has been changed to "Phytoseiidae".
[Revised text: “Phytoseiidae”]
Comment 2:
Page 12, line 275: The abbreviation “sp.” should not be italicized. Please ensure this formatting is applied consistently throughout the manuscript.
Response 2:
We agree with this comment. The abbreviation “sp.” has been corrected to the non-italicized form at Page 12, line 275, and this correction has been applied consistently throughout the manuscript to ensure uniform formatting.
[Revised text: “sp.” in non-italicized form applied consistently throughout the manuscript]